# Association of functional health literacy and cognitive ability with self-reported diabetes in the English Longitudinal Study of Ageing: a prospective cohort study

Chloe Fawns-Ritchie ,[1] Jackie Price,[2] Ian J Deary[1]

[1]Department of Psychology, The University of Edinburgh, Edinburgh, UK
[2]Usher Institute, The University of Edinburgh, Edinburgh, UK

**Correspondence to**
Dr Chloe Fawns-Ritchie;
c.fawns-ritchie@ed.ac.uk

## ABSTRACT

**Objectives** We investigated whether functional health literacy and cognitive ability were associated with self-reported diabetes.

**Design** Prospective cohort study.

**Setting** Data were from waves 2 (2004–2005) to 7 (2014–2015) of the English Longitudinal Study of Ageing (ELSA), a cohort study designed to be representative of adults aged 50 years and older living in England.

**Participants** 8669 ELSA participants (mean age=66.7, SD=9.7) who completed a brief functional health literacy test assessing health-related reading comprehension, and 4 cognitive tests assessing declarative memory, processing speed and executive function at wave 2.

**Primary outcome measure** Self-reported doctor diagnosis of diabetes.

**Results** Logistic regression was used to examine cross-sectional (wave 2) associations of functional health literacy and cognitive ability with diabetes status. Adequate (compared with limited) functional health literacy (OR 0.71, 95% CI 0.61 to 0.84) and higher cognitive ability (OR per 1 SD=0.73, 95% CI 0.67 to 0.80) were associated with lower odds of self-reporting diabetes at wave 2. Cox regression was used to test the associations of functional health literacy and cognitive ability measured at wave 2 with self-reporting diabetes over a median of 9.5 years follow-up (n=6961). Adequate functional health literacy (HR 0.64; 95% CI 0.53 to 0.77) and higher cognitive ability (HR 0.77, 95% CI 0.69 to 0.85) at wave 2 were associated with lower risk of self-reporting diabetes during follow-up. When both functional health literacy and cognitive ability were added to the same model, these associations were slightly attenuated. Additionally adjusting for health behaviours and body mass index fully attenuated cross-sectional associations between functional health literacy and cognitive ability with diabetes status, and partly attenuated associations between functional health literacy and cognitive ability with self-reporting diabetes during follow-up.

**Conclusions** Adequate functional health literacy and better cognitive ability were independently associated with lower likelihood of reporting diabetes.

## STRENGTHS AND LIMITATIONS OF THIS STUDY

⇒ This study used data from the English Longitudinal Study of Ageing, a large prospective cohort study designed to be representative of community-dwelling adults aged over 50 years living in England.
⇒ Participants were followed up for a median of 9.5 years to determine whether they were diagnosed with diabetes.
⇒ Diabetes status was self-reported.
⇒ Health literacy and cognitive ability assessments were brief.

## INTRODUCTION

Diabetes is a common chronic condition in older adulthood and is associated with substantial morbidity and mortality.[1] Type 2 diabetes, the most common type of diabetes, is at least partly preventable.[1] Understanding the characteristics of those most at risk of developing diabetes is important for appropriately targeting diabetes education and interventions. Risk factors for developing diabetes include older age, deprivation and obesity.[1]

Lower cognitive ability may be a risk factor for diabetes. Cognitive ability can be conceptualised as a composite term for a range of different but overlapping mental capabilities, including the ability to learn, plan, problem solve and process information.[2] Cognitive ability is closely related to but distinct from educational attainment and correlations between cognitive ability and education range from 0.40 to 0.80.[3] This general mental capability has been found to be associated with many different aspects of health.[2] Studies examining the association between cognitive ability and diabetes have found mixed results. One study[4] found that childhood cognitive ability did not predict diabetes in midlife

when individually adjusting for a range of demographic variables including education. Others have found that lower cognitive ability in early life was associated with higher risk of diabetes in adulthood.[5][6] Whereas the first study[5] did not adjust for educational attainment or measures of socioeconomic status, the latter[6] found that individuals with lower cognitive ability in early adulthood had higher rates of diabetes in midlife, even after adjusting for education and indicators of socioeconomic status. Individuals with higher cognitive ability might have the cognitive skills required to self-manage their health, take better care of themselves throughout life, and thus reduce the risk of developing diabetes.[2][5]

Health literacy is the 'capacity to obtain, process and understand basic health information and services needed to make basic health decisions',[7] and it might also play a role in diabetes. Health literacy is a multifaceted construct thought to encompass all of the skills required to make decisions about one's health, including the ability to access, appraise and apply health information.[8][9] One component of health literacy is functional health literacy—the reading, writing and numeracy skills needed to understand basic health information.[10] These skills are thought to be required, for example, to understand and correctly follow the instructions on a packet of prescription medication. In cross-sectional studies, rates of diabetes are higher in those with low functional health literacy, even after adjusting for age, sex, income and education.[11][12] In one study, participants with inadequate functional health literacy were 48% more likely to report having diabetes when compared with participants with adequate health literacy, adjusting for sociodemographic and health variables.[12] Associations between health literacy and diabetes may differ by sex. Women with low health literacy were found to be more than twice as likely to have diabetes compared with those with high literacy after adjusting for age, race, income, education, body mass index (BMI) and smoking and alcohol status, however, health literacy was not associated with diabetes status in men.[13] Individuals with lower functional health literacy—at least in women—might lack the health-related reading and writing skills required to obtain, understand and follow health advice, such as eating well and exercising, which might reduce the risk of diabetes.[7]

In patients with diabetes, higher functional health literacy has consistently been associated with greater diabetes knowledge.[14–16] A very small association between higher functional health literacy and lower glycated haemoglobin ($HbA_{1C}$) levels in patients with diabetes has been reported in a meta-analysis of 26 studies ($r=-0.048$, $p=0.027$).[15] Whereas studies have investigated the association between functional health literacy and disease management in people with diabetes, little is known about whether functional health literacy is associated with risk of developing diabetes.

Functional health literacy and cognitive ability test scores are positively correlated.[17–19] Rank-order correlations between general cognitive ability and three functional health literacy tests ranged from 0.37 to 0.50.[18] Researchers have sought to determine the role of cognitive ability in the association between functional health literacy and a range of health outcomes. Most (but not all)[20] studies have found that cognitive ability partly or entirely attenuates the association between functional health literacy and health.[21–24] One study[19] sought to determine whether health literacy and cognitive ability had independent associations with performance on various health-related tasks, including comprehending written and video-presented health information and using health-related props, such as a pill bottle. Using three different measures of functional health literacy, the association between functional health literacy and performance on the health-related tasks were attenuated by between 70.6% and 77.7% when including cognitive ability in the same model compared with models not including cognitive ability.[19] Any association between functional health literacy and diabetes may be attenuated when also measuring cognitive ability.

The aim of the current study was to better understand the associations of functional health literacy and cognitive ability with diabetes. Using data from the English Longitudinal Study of Ageing (ELSA),[25] this study investigated whether functional health literacy and cognitive ability were independently associated with diabetes. First, the cross-sectional associations between functional health literacy, cognitive ability and self-reported diabetes were investigated. Second, participants without diabetes at baseline were followed-up for up to 10 years to determine whether functional health literacy and cognitive ability were independently associated with newly reporting diabetes during the follow-up.

## METHODS

### Participants

This study used data from core members of the ELSA study, a prospective cohort study of community-dwelling adults residing in England. ELSA was designed to be representative of adults aged 50 years and older living in England.[25] The wave 1 (2002–2003) sample consisted of 11 391 participants who had previously participated in the Health Survey for England between 1998 and 2001, who were born before 1 March 1952, and who were living in a private household in England.[25] ELSA participants have been followed up every 2 years and the sample has been refreshed at waves 3, 4, 6 and 7 to ensure the sample is representative of adults aged over 50 years. This study used data from waves 2 (2004–05) to 7 (2014–15), and baseline, here, was considered to be wave 2 (n=8726), which was when the functional health literacy assessment was introduced.

At each wave, a face-to-face interview was used to measure topics including health, lifestyle and economic circumstances. Face-to-face interviews were carried out in the participant's own home using computer-assisted interviewing. Participants answered a self-completion

questionnaire including questions about diet and alcohol consumption. A nurse interview was carried out at waves 2, 4 and 6 to assess physical measurements including height and weight, and blood and saliva samples were taken to measure biomarkers of disease. Detailed descriptions of the sample design and data collected in ELSA are reported elsewhere.[25]

### Patient and public involvement

Participants were not involved in the development of any part of this study.

### Measures

#### Diabetes

*Baseline diabetes status*

Individuals who answered 'yes' to 'Has a doctor ever told you that you have diabetes?' at wave 2 were categorised as having diabetes. This question did not differentiate which type of diabetes the participant was diagnosed with.

*Diabetes during follow-up*

This analysis was restricted to participants who did not self-report diabetes at wave 2 and who had at least one wave of follow-up between waves 3 and 7. Participants who did not self-report diabetes at wave 2 and who subsequently answered 'yes' to 'Has a doctor ever told you that you have diabetes?' any time between waves 3 and 7 were categorised as having newly diagnosed diabetes during follow-up. As all participants were aged over 50 years at diagnosis, these cases are probably cases of type 2 diabetes.

*Date of diabetes diagnosis*

Individuals who self-reported diabetes were asked which month and year they were diagnosed. Date of diabetes diagnosis was used to calculate the time between wave 2 assessment and diabetes diagnosis.

#### Functional health literacy

A four-item functional health literacy test taken from the Adult Literacy and Life Skills Survey,[26] and the International Adult Literacy Survey[27] was administered during the wave 2 interview. This test assessed health-related reading comprehension skills which are thought to be required to successfully understand written materials commonly encountered in healthcare. Participants were presented with a piece of paper containing a label for a packet of over-the-counter medication. Participants were asked four questions about the information on this label (eg, 'what is the maximum number of days you may take this medicine?'). The score was the number of correctly answered questions. As has been done in other studies,[28 29] performance was categorised as adequate (4/4 correct) or limited (<4 correct).

#### Cognitive ability

Scores on different cognitive tests tend to be positively correlated.[30] Data reduction techniques such as principal component analysis (PCA) are often used to capture the covariance among a range of difference cognitive tests. This shared variance can then be used as a measure of general cognitive ability.[31] Four tests administered during the wave 2 interview that are designed to assess cognitive domains that decline with increasing age[32] were entered into a PCA to create a measure general cognitive ability.

Word list learning tests, in which participants are required to remember a list of words immediately and then after a delay are commonly used to assess verbal declarative memory and learning.[33] Here, the immediate and delayed word recall tests were used. Participants were read a list of 10 words and were asked to immediately recall as many of the words as possible. The score was the number of words recalled immediately. After a short delay, in which the words were not repeated, participants were asked to remember the 10 words again. The score was the number of words recalled after a delay. Verbal fluency tests, in which participants are asked to produce as many words as possible in a set time following a set of rules, are often used to measure executive function.[33] Category fluency was used to assess executive function in ELSA. Participants were instructed to name as many animals as possible. The score was the number of animals named in 60 s. Tests of processing speed involve completing a simple task as quickly as possible and common tests include using a code to write as many symbols as possible, or finding symbols among distractors and scoring them out as quickly as possible.[33 34] Letter cancellation was used to assess processing speed. Participants were presented with a piece of paper containing letters of the alphabet arranged in rows and columns. The task was to scan the piece of paper and score out all Ps and Ws. The score was the combined number of Ps and Ws scored out in 60 s.

Scores of 0 on animal fluency (n=48) and letter cancellation (n=3) were removed as scores of 0 on these tests suggest participants either did not complete the task or did not understand the task. Scores of ≥50 on animal fluency (n=4), and ≥60 on the letter cancellation (n=3) were removed as these scores were extremely high given the 60 s time limit for these tests and these values are greater than 4 SDs from the mean.

We did not include tests of self-reported memory, prospective memory or orientation in time in the measure of general cognitive ability. Self-reported memory was not included because this is a subjective test. Prospective memory was not included because the test consists of only one trial. Orientation in time is a four-item test in which participants are asked to recall the date. It has limited variance and is most frequently used as a brief screening tool for cognitive impairment.

Only the first principal component had an eigenvalue >1. The scree plot also indicated one component. Scores from the first principal component were saved and used as a measure of cognitive ability (mean=0.00, SD=1.00). The first component accounted for 57% of the variance in the scores on the four cognitive tests. The loadings were: Immediate word recall=0.83,

delayed word recall=0.85, animal fluency=0.72 and letter cancellation=0.58.

## Covariates

Age (in years), sex, BMI, health behaviours, number of cardiovascular comorbidities and indicators of socioeconomic status were used as covariates. Unless otherwise stated, all were self-reported at the wave 2 interview. Prior to releasing data, ELSA set the age of all participants aged over 90 years to 90 years to reduce the risk of disclosure. Participants were asked whether they smoked cigarettes nowadays and were categorised as current smokers or non-smokers. Participants were asked how often they took part in moderate and vigorous physical activity (more than once a week, once a week, one to three times a month, and hardly ever/never). Physical activity levels were categorised as vigorous activity at least once per week, moderate activity at least once per week, and physically inactive. Participants were asked about their frequency of alcohol consumption in the past 12 months in the self-completion questionnaire. This was categorised as never, rarely, at least once a month, at least once a week and daily/almost daily. Height and weight, measured during the wave 2 nurse interview, were used to calculate BMI ($kg/m^2$). Cardiovascular comorbidities were assessed by counting the number of self-reported cardiovascular conditions from hypertension, angina, heart attack, heart murmur, abnormal heart rhythm, stroke and high cholesterol. Age that participants left full-time education was categorised as: age 14 or under, 15–16 years, 17–18 years and age 19 or older. Social class was categorised using the National Statistics Socioeconomic Classification 3 categories[35]; managerial and professional, intermediate and routine and manual.

## Analysis

All analyses was performed in R. Independent t-tests were used to compare those with and without diabetes at wave 2 and those who did and did not self-report diabetes at follow-up on normally distributed continuous variables. Mann-Whitney U tests were used for non-normal continuous variables, and $\chi^2$ tests were used for categorical variables. Spearman rank-order correlations were calculated between all predictor variables and covariables.

Binary logistic regression was used to test the cross-sectional association of functional health literacy and cognitive ability with diabetes reported at wave 2. Cox regression was used to investigate whether functional health literacy and cognitive ability test scores at wave 2 were associated with newly reported diabetes between waves 2 and 7. In the Cox regression analysis, time-to-event was taken as the difference, in days, between date of wave 2 interview and date of diabetes diagnosis for those who self-reported diabetes. For other participants, time-to-event was the difference between date of wave 2 interview and the date of last interview. Month and year, but not day, were recorded for date of interview and date

of diabetes diagnosis. To create a date variable (yyyy.mm.dd), the day was set to the middle of the month.

For the logistic regressions and Cox regressions, 7 models were run. Age and sex were entered into all models. Functional health literacy and cognitive ability were entered individually in models 1 and 2, respectively. Both functional health literacy and cognitive ability were added in model 3 to determine whether the size of the functional health literacy-diabetes and cognitive ability-diabetes associations changed when simultaneously entering both these variables. Functional health literacy and cognitive ability were also entered together in models 4–7. To assess whether BMI and health behaviours accounted for these associations, BMI, smoking status, alcohol consumption and physical activity were added in model 4. Diabetes is a risk factor for cardiovascular disease.[36] Associations between poorer cognitive ability and cardiovascular disease are also well established.[37 38] It is possible that any association between functional health literacy and cognitive ability with diabetes may be because of these associations with cardiovascular disease. To determine whether any association between functional health literacy and cognitive ability with diabetes was attenuated when adjusting for cardiovascular disease, number of cardiovascular comorbidities was added in model 5. Age of leaving full-time education and occupational social class were added in model 6. A fully-adjusted model (model 7) adjusted for functional health literacy, cognitive ability and all covariates.

This study was interested in the associations of functional health literacy and cognitive ability with self-reported diabetes and the independence of these associations with respect to other health and socioeconomic-related variables. In the main text, we report the ORs and HRs for functional health literacy and cognitive ability only. The estimates for all variables entered into the models are reported in online supplemental materials.

## RESULTS

Of the 8726 ELSA participants who completed wave 2, 3 participants were removed who answered 'don't know' to whether a doctor had diagnosed them with diabetes. A further 54 participants were removed because they selected that they had 'diabetes or high blood sugar' from a Showcard listing cardiovascular conditions, but when asked whether a doctor had ever told them they had diabetes, they answered 'no'. The analytical sample consisted of 8669 participants. Participant characteristics are reported in table 1.

### Baseline diabetes status

At baseline, 708 (8.2%) participants self-reported a diagnosis of diabetes. Compared with those without diabetes, those with diabetes were more likely to have limited functional health literacy (42.2% vs 32.3%) and have lower cognitive ability (diabetes mean=−0.36, SD=0.97; no diabetes mean=0.03, SD=1.00; Cohen's

**Table 1** Participant characteristics by diabetes status

| | | Diabetes reported at wave 2 | | | | Diabetes reported during follow-up* | | |
| --- | --- | --- | --- | --- | --- | --- | --- | --- |
| | n | No diabetes (n=7961) | Diabetes (n=708) | P value | n | No diabetes (n=6455) | Diabetes (n=506) | P value |
| Age, mean (SD) | 8669 | 66.46 (9.70) | 69.38 (9.16) | <0.001 | 6961 | 66.02 (9.53) | 65.51 (8.59) | <0.001 |
| Sex, n (%) | 8669 | | | <0.001 | 6961 | | | <0.001 |
| Male | | 3522 (44.2) | 379 (53.5) | | | 2791 (43.2) | 262 (51.8) | |
| Female | | 4439 (55.8) | 329 (46.5) | | | 3664 (56.8) | 244 (48.2) | |
| Age left full-time education, n (%) | 8468 | | | <0.001 | 6809 | | | <0.001 |
| ≤14 years | | 1641 (21.1) | 210 (30.6) | | | 1222 (19.3) | 107 (21.8) | |
| 15–16 years | | 4085 (52.5) | 349 (50.8) | | | 3283 (52.0) | 302 (61.6) | |
| 17–18 years | | 1009 (13.0) | 55 (8.0) | | | 870 (13.8) | 45 (9.2) | |
| ≥19 years | | 1046 (13.4) | 73 (10.6) | | | 944 (14.9) | 36 (7.3) | |
| Social class, n (%) | 8508 | | | <0.001 | 6846 | | | <0.001 |
| Managerial and professional | | 2444 (31.2) | 194 (28.4) | | | 2067 (32.6) | 133 (26.7) | |
| Intermediate | | 1979 (25.3) | 131 (19.2) | | | 1662 (26.2) | 104 (20.9) | |
| Routine and manual | | 3403 (43.5) | 357 (52.3) | | | 2619 (41.3) | 261 (52.4) | |
| Health literacy, n (%) | 8293 | | | <0.001 | 6736 | | | <0.001 |
| Adequate | | 5172 (67.7) | 376 (57.8) | | | 4351 (69.7) | 300 (61.2) | |
| Limited | | 2471 (32.3) | 274 (42.2) | | | 1895 (30.3) | 190 (38.8) | |
| Cognitive ability, mean (SD) | 8335 | 0.03 (1.00) | −0.36 (0.97) | <0.001 | 6746 | 0.10 (0.98) | −0.04 (0.89) | <0.001 |
| BMI, mean (SD) | 7179 | 27.71 (4.79) | 30.45 (5.37) | <0.001 | 5997 | 27.46 (4.64) | 31.21 (5.28) | <0.001 |
| Current smoker, n (%) | 8622 | | | 0.377 | 6929 | | | <0.001 |
| Yes | | 1216 (15.4) | 99 (14.1) | | | 934 (14.5) | 105 (20.8) | |
| No | | 6704 (84.6) | 603 (85.9) | | | 5490 (85.5) | 400 (79.2) | |
| Alcohol, n (%) | 7577 | | | <0.001 | 6239 | | | <0.001 |
| Never | | 723 (10.3) | 112 (19.3) | | | 565 (9.7) | 49 (11.2) | |
| Rarely | | 1076 (15.4) | 124 (21.3) | | | 863 (14.9) | 90 (20.6) | |
| At least once a month | | 827 (11.8) | 85 (14.6) | | | 669 (11.5) | 70 (16.1) | |
| At least once a week | | 2662 (38.1) | 171 (29.4) | | | 2255 (38.9) | 149 (34.2) | |
| Daily/almost daily | | 1708 (24.4) | 89 (15.3) | | | 1451 (25.0) | 78 (17.9) | |
| Physical activity, n (%) | 8665 | | | <0.001 | 6958 | | | <0.001 |
| Vigorous activity | | 2236 (28.1) | 108 (15.2) | | | 1938 (30.0) | 116 (22.9) | |
| Moderate activity | | 3888 (48.9) | 305 (43.1) | | | 3194 (49.5) | 233 (46.0) | |
| Inactive | | 1833 (23.0) | 295 (41.7) | | | 1320 (20.5) | 157 (31.0) | |
| Number of cardiovascular comorbidities, mean (SD) | 8669 | 0.67 (0.91) | 1.28 (1.13) | <0.001 | 6961 | 0.64 (0.88) | 0.89 (1.04) | <0.001 |

*Diabetes reported at follow-up comparisons are based on a subsample of participants who did not self-report diabetes at wave 2 and with at least one wave of follow-up.
BMI, body mass index.

d=0.40). Participants with diabetes were older (diabetes mean=69.36, SD=9.16; no diabetes mean=66.46, SD=9.70) and more likely to be male (53.5% vs 44.2%) than those without. Those with diabetes were also more likely to leave full-time education at a younger age, be from a less professional social class, have a higher BMI, consume

less alcohol, be inactive and self-report more cardiovascular comorbidities (table 1). Rank-order correlations between predictor variables and co-variables are reported in table 2. Adequate functional health literacy was moderately correlated with higher cognitive ability (r=0.31, p<0.001).

ORs and 95% CIs for the associations between functional health literacy and cognitive ability with self-reported diabetes at wave 2 are reported in table 3 and online supplemental table S1. Box-Tidwell tests were performed whereby an interaction term between each continuous predictor variable and the log of that variable were added to the model to check the assumption that there is a linear relationship between each continuous predictor and the logit of the outcome. The interaction between age and log(age) and the interaction between number of cardiovascular comorbidities and log(number of cardiovascular comorbidities) was significant. Therefore, the assumptions of the linearity of the logit was violated. To overcome this, an age-squared term was included in all models, and a squared term for number of cardiovascular comorbidities was included in models 5 and 7.

Participants with adequate functional health literacy were 29% less likely to self-report diabetes (model 1 OR 0.71; 95% CI 0.61 to 0.84). A 1 SD higher cognitive ability was associated with 27% lower odds of self-reported diabetes (model 2 OR 0.73; 95% CI 0.67 to 0.80). The association between functional health literacy and diabetes was attenuated by 38% (OR 0.82; 95% CI 0.69 to 0.98) and the association between cognitive ability and diabetes was attenuated by 19% (OR 0.78; 95% CI 0.70 to 0.86) when entering both functional health literacy and cognitive ability in model 3. Both remained significantly associated with diabetes.

BMI and health behaviours were added in model 4. The associations between functional health literacy and cognitive ability with diabetes were attenuated and no longer significant. The cognitive ability-diabetes association was not attenuated after adjusting for cardiovascular comorbidities (model 5) or when adjusting for education and social class (model 6). Cognitive ability remained significantly associated with diabetes in these models. The association between functional health literacy and diabetes was slightly attenuated and no longer significant when adjusting for cardiovascular comorbidities (model 5) and education and social class (model 6). In the fully adjusted model (model 7), the size of the associations between functional health literacy and cognitive ability with diabetes were reduced further and were non-significant.

In the fully-adjusted model (model 7; online supplemental table S1) older age, male sex, having a higher BMI and reporting more cardiovascular comorbidities were associated with higher odds of having diabetes. The association between number of cardiovascular comorbidities and diabetes became less strong as the number of comorbidities increased. Those who reported drinking alcohol at least once per month, rarely, or who never drank

alcohol in the last 12 months were more likely to self-report diabetes when compared with those who reported drinking daily/almost daily. Compared with those who reported being physically inactive, those who took part in moderate or vigorous physical activity at least once per week were less likely to self-report diabetes.

## Diabetes during follow-up

Of the 7961 participants who did not self-report diabetes at wave 2, 6961 participants had at least one wave of follow-up between waves 3 and 7. They form the analytic sample for the association between functional health literacy, cognitive ability and self-reported diabetes during follow-up. A total of 506 (7.3%) participants reported a new diagnosis of diabetes between wave 3 and wave 7, whereas 6455 (92.7%) participants did not. Median time to follow-up was 9.5 years. Mean time to censor was 4.7 years (SD=3.1) for those with diabetes and 7.8 years (SD=2.9) for those without. Participant characteristics are reported in table 1. Compared with participants who did not self-report diabetes during follow-up, those who did were more likely to have limited functional health literacy (38.8% vs 30.3%) and had lower cognitive ability (diabetes mean=−0.04, SD=0.89; no diabetes mean=0.10, SD=0.98, Cohen's d=0.15) at wave 2. Participants who reported diabetes were younger (diabetes mean=65.51, SD=8.59; no diabetes mean=66.02; SD=9.53) and more likely to be male (51.8% vs 43.2%) than those without. Compared with those without diabetes, participants who reported diabetes during follow-up were more likely to have left full-time education at a younger age, be from a less professional social class, smoke, consume less alcohol, be inactive, and to report more cardiovascular comorbidities at wave 2 (table 1).

The HRs and 95% CIs for the association between functional health literacy, cognitive ability and self-reporting diabetes during follow-up are reported in table 4 and online supplemental table S2. Adequate functional health literacy at wave 2 was associated with a 36% lower risk of reporting diabetes (model 1 HR 0.64; 95% CI 0.53 to 0.77). A 1 SD higher cognitive ability at wave 2 was associated with a 23% lower risk of reporting diabetes (model 2 HR 0.77; 95% CI 0.69 to 0.85). The association between functional health literacy and diabetes was attenuated by 22% after adjustment for cognitive ability (model 3 HR 0.72; 95% CI 0.59 to 0.87), and the association between cognitive ability and diabetes was attenuated by 9% after adjusting for functional health literacy (HR 0.79; 95% CI 0.71 to 0.88). Both functional health literacy and cognitive ability remained significant predictors of reporting diabetes during the follow-up.

BMI and health behaviours were added in model 4. The associations of functional health literacy and cognitive ability with reporting diabetes were further attenuated but remained statistically significant. When adjusting for number of cardiovascular comorbidities, the association between functional health literacy and cognitive ability with diabetes remained almost unchanged (model

**Table 2** Spearman rank-order correlations between predictor variables and co-variables (n=6463–8660)

| | Age | Sex | Education | Social class | Health literacy | Cognitive ability | BMI | Smoking | Alcohol | Physical activity | CV comorbid |
|---|---|---|---|---|---|---|---|---|---|---|---|
| Age | | | | | | | | | | | |
| Sex | −0.03** | | | | | | | | | | |
| Education | −0.41*** | 0.02 | | | | | | | | | |
| Social class | 0.08*** | −0.09*** | −0.41*** | | | | | | | | |
| Health literacy | −0.16*** | 0.01 | 0.23*** | −0.18*** | | | | | | | |
| Cognitive ability | −0.47*** | −0.09*** | 0.39*** | −0.27*** | 0.31*** | | | | | | |
| BMI | −0.07*** | 0.02 | −0.06*** | 0.08*** | −0.04** | −0.01 | | | | | |
| Smoking | −0.13*** | 0.01 | −0.05*** | 0.12*** | −0.04*** | −0.02 | −0.09*** | | | | |
| Alcohol | −0.11*** | 0.21*** | 0.22*** | −0.20*** | 0.09*** | 0.16*** | −0.11*** | −0.04*** | | | |
| Physical activity | −0.26*** | 0.10*** | 0.23*** | −0.15*** | 0.14*** | 0.26*** | −0.11*** | −0.09*** | 0.18*** | | |
| CV comorbid | 0.18*** | 0.00 | −0.11*** | 0.05*** | −0.06*** | −0.11*** | 0.14*** | −0.03* | −0.08*** | −0.14*** | |

Sex is coded 0 for female,1 for male; Education is age of leaving full-time education and is coded 1 for age 14 years or less, 2 for age 15–16 years, 3 for age 17–18 years and 4 for 19 years or older; Social class is coded 1 for managerial and professional, 2 for intermediate and 3 for routine and manual; Health literacy is coded 0 for limited and 1 for adequate; Smoking is coded 0 for current non-smoker and 1 for current smoker; alcohol is the frequency of alcohol consumed in the last 12 months and is coded 0 for never, 1 for rarely, 2 for at least once a month, 3 for at least once a week, 4 for daily/almost daily; Physical activity is coded 0 for inactive, 1 for moderate activity at least once per week, 2 for vigorous activity at least once per week; CV comorbid is the number of CV comorbidities self-reported from hypertension, angina, heart attack, heart failure, heart murmur, abnormal heart rhythm, stroke and high cholesterol.

*P<0.05, **p<0.01, ***p<0.001.

BMI, body mass index; CV, cardiovascular.

**Table 3** OR (95% CIs) from logistic regression models of the association between functional health literacy and cognitive ability with self-reported diabetes at wave 2

| | Model 1: Health literacy | Model 2: Cognitive ability | Model 3: Health literacy and cognitive ability | Model 4: +BMI and health behaviours | Model 5: +CV comorbidities | Model 6: +Education and social class | Model 7: Fully adjusted |
|---|---|---|---|---|---|---|---|
| Adequate health literacy | 0.71*** (0.61 to 0.84) | – | 0.82* (0.69 to 0.98) | 0.97 (0.78 to 1.21) | 0.85 (0.72 to 1.02) | 0.84 (0.70 to 1.01) | 0.98 (0.78 to 1.23) |
| Cognitive ability | – | 0.73*** (0.67 to 0.80) | 0.78*** (0.70 to 0.86) | 0.90 (0.80 to 1.02) | 0.78*** (0.71 to 0.87) | 0.78*** (0.71 to 0.87) | 0.87 (0.76 to 1.00) |

All models adjusted for age, age-squared, and sex. Model 1 n=8293, Model 2 n=8335, Model 3 n=8185. Model 4 (n=6302) adjusted for body mass index, frequency of alcohol consumption in the past 12 months and physical activity. Model 5 (n=8185) adjusted for number of CV comorbidities reported, and a squared term number of CV comorbidities reported. Model 6 (n=7861) adjusted for age left full-time education, and occupational social class. Model 7 (n=6086) adjusted for all covariates.
*P<0.05, **p<0.01, ***p<0.001.
BMI, body mass index; CV, cardiovascular.

**Table 4** HRs (95% CIs) from Cox regression models of the association between functional health literacy and cognitive ability with self-reporting diabetes during follow-up

| | Model 1: Health literacy | Model 2: Cognitive ability | Model 3: Health literacy and cognitive ability | Model 4: +BMI and health behaviours | Model 5: +CV comorbidities | Model 6: +Education and social class | Model 7: Fully adjusted |
|---|---|---|---|---|---|---|---|
| Adequate health literacy | 0.64*** (0.53 to 0.77) | – | 0.72** (0.59 to 0.87) | 0.79* (0.64 to 0.99) | 0.73** (0.60 to 0.88) | 0.79* (0.65 to 0.97) | 0.85 (0.68 to 1.06) |
| Cognitive ability | – | 0.77*** (0.69 to 0.85) | 0.79*** (0.71 to 0.88) | 0.85* (0.74 to 0.96) | 0.80*** (0.71 to 0.89) | 0.84** (0.75 to 0.95) | 0.88 (0.77 to 1.01) |

All models adjusted for age and sex. Models 1 (n=6736) had 490 diabetes events, model 2 (n=6746) had 491 diabetes events, model 3 (n=6654) had 484 diabetes events. Model 4 (n=5357; 377 diabetes events) adjusted for body mass index, frequency of alcohol consumption in the past 12 months, and physical activity. Model 5 (n=6654; 484 diabetes events) adjusted for number of CV comorbidities reported. Model 6 (n=6409; 492 diabetes events) adjusted for age left full-time education and occupational social class. Model 7 (n=5186, 360 diabetes events) adjusted for all covariates.
*P<0.05, **p<0.01, ***p<0.001.
BMI, body mass index; CV, cardiovascular.

5) and both remained significantly associated with diabetes. Education and social class was added in model 6. The size of the association between functional health literacy and cognitive ability with diabetes were slightly reduced but remained statistically significant. In the fully-adjusted model (model 7) the associations between functional health literacy and cognitive ability and reporting diabetes were further reduced and no longer significant.

In the fully-adjusted model (model 7; online supplemental table S2) male participants, those with a higher BMI, current smokers, and those who reported consuming alcohol rarely (compared with daily/almost daily) at wave 2 were more likely to report diabetes during follow-up. Participants who reported leaving education at age 19 years or older were less likely to report diabetes during follow-up compared with those who left at age 14 years or younger.

### Sensitivity analysis
#### Missing data
There was missing data. For the cross-sectional analyses, 70% of participants had complete data. For the longitudinal analyses, 75% of participants had complete data. All models were rerun using only participants with complete data on all variables. These results are reported in online supplemental tables 3 and 4. The pattern of associations were generally similar; however, the sizes of the associations tended to be slightly weaker compared with the full sample. For the cross-sectional analysis, functional health literacy was no longer significantly associated with diabetes status in model 3 when adjusting for functional health literacy and cognitive ability (online supplemental table S3). For the longitudinal analysis, when adjusting for BMI and health behaviours (model 4; online supplemental table S4), functional health literacy was no longer associated with reporting diabetes during follow-up.

#### Undiagnosed diabetes
It is possible that some participants not reporting diabetes may have undiagnosed diabetes. To identify participants who may have undiagnosed diabetes $HbA_{1c}$ levels collected by blood draw during the nurse interview (waves 2, 4 and 6) were used.[25] Participants who did not report diabetes but who had $HbA_{1c}$ levels of ≥47.5 mmol/mol (6.5%) were categorised as having suspected undiagnosed diabetes. The models were rerun after removing these individuals to determine whether the results differ from those reported in the main models.

A total of 5783 participants who formed the analytical sample for the cross-sectional analysis had $HbA_{1c}$ levels available from the wave 2 nurse interview (399 self-reporting diabetes; 5384 not self-reporting diabetes). Of the 5384 participants who did not self-report diabetes at wave 2 and who had $HbA_{1c}$ levels available at wave 2, 112 (2.1%) participants had $HbA_{1c}$ levels of ≥47.5 mmol/mol (6.5%). Models were rerun on this subsample after removal of these 112 participants with suspected undiagnosed diabetes (n=5671). The results are reported in

online supplemental table S5. The associations between cognitive ability and diabetes status at wave 2 are very similar to those reported in the main model. Using this subsample, the size of the associations between functional health literacy and diabetes were reduced and were no longer significant in model 1 (functional health literacy only; online supplemental table S5) and model 3 (functional health literacy and cognitive ability; online supplemental table S5).

The Cox regressions were also rerun after removal of participants with suspected undiagnosed diabetes. The follow-up period was restricted to waves 3–6 (mean follow-up=7.5 years), as $HbA_{1c}$ levels were not available at wave 7. A total of 4425 participants who formed the analytical sample for the Cox models had $HbA_{1c}$ levels collected at wave 4 and/or wave 6 (279 self-reporting diabetes between waves 3 and 6; 4146 not self-reporting diabetes during follow-up). A total of 147 participants who reported not having diabetes at waves 3 and 4 had $HbA_{1c}$ levels of ≥47.5 mmol/mol (6.5%) at wave 4 and were removed. A further 72 participants reported not having diabetes between waves 3 and 6 but had $HbA_{1c}$ levels of ≥47.5 mmol/mol (6.5%) at wave 6 and were removed. The Cox regression models were re-run on this sample (n=4206; 212 reporting diabetes during follow-up; 3994 not reporting diabetes during the follow-up). The results are reported in online supplemental table S6. The size of the associations between limited functional health literacy and self-reporting diabetes during follow-up became even stronger. In the fully-adjusted model (model 7, online supplemental table S6), the association between limited functional health literacy and diabetes remained significant. For cognitive ability, the strength of the associations were generally similar to the main models. However, after adjusting for BMI and health behaviours (model 4, online supplemental table S6) the size of the association between cognitive ability and diabetes was slightly attenuated and no longer significant.

### DISCUSSION
Using a sample of middle-aged and older adults living in England, this study found that adequate functional health literacy and better cognitive ability were associated with lower odds of self-reporting diabetes. These associations were attenuated when functional health literacy and cognitive ability were entered in the same model, though both independently contributed to diabetes. These associations were further attenuated and non-significant when adjusting for BMI and health behaviours. Adjusting for cardiovascular comorbidities and indicators of socioeconomic status did not attenuate the association between cognitive ability and diabetes, however, for functional health literacy there was a small attenuation and these associations were no longer significant. When adjusting for all covariates simultaneously, neither functional health literacy nor cognitive ability was associated with diabetes at wave 2.

Adequate health literacy and better cognitive ability, measured at wave 2, were associated with reduced risk of self-reporting diabetes during a median of 9.5 years follow-up. Both functional health literacy and cognitive ability were independently associated with self-reported diabetes when both were entered in the same model. These associations remained when separately adjusting for BMI and health behaviours, cardiovascular comorbidities and education and social class. However, neither health literacy nor cognitive ability were associated with reporting diabetes during follow-up when all covariates were entered together.

Previous cross-sectional studies have found that individuals with lower functional health literacy are more likely to report having diabetes[11 12] and longitudinal studies have found that that lower cognitive ability earlier in life is associated with an increased risk of diabetes.[5 6] This study is the first longitudinal study to examine whether functional health literacy is associated with self-reporting a new diagnosis of diabetes, and the first to examine whether cognitive ability and functional health literacy have independent associations with diabetes. The results reported here suggest that cognitive capabilities and health-related reading comprehension skills, though related, contribute independently to diabetes.

There are obvious similarities between tests of cognitive ability and functional health literacy. The Rapid Estimate of Adult Literacy in Medicine (REALM)[39] is a popular health literacy test which involves the ability to read and pronounce health-related words of varying complexity. More ecologically valid assessments of functional health literacy such as the Test of Functional Health Literacy in Adults (TOFHLA)[10] and the health literacy test used in the current study involve participants using mock health-related props, such as prescription labels or a medical appointment slips, and answering questions about the information presented. Successful completion of these tests will require the ability to process information, plan and problem solve (ie, cognitive ability).[2]

Some have suggested that functional health literacy variance is mostly overlapping with cognitive ability.[23 40] If this were true, one would expect the association between functional health literacy and diabetes to be fully attenuated when adjusting for cognitive ability. This is not what was found here. Only some of the association of functional health literacy and diabetes was accounted for by cognitive ability. The level of independence between health literacy and cognitive ability may vary depending on the assessments used to measure health literacy and cognitive ability.[22] The cognitive ability measure used here included four brief cognitive ability tests that assessed memory, executive function and processing speed, and did not include other important domains of cognitive function, such as reasoning, that are known to load highly on general cognitive ability.[41] The health literacy assessment was also very brief. Some of the unique contribution of functional health literacy might be residual cognitive capability that was not picked up by the relatively brief

measures of cognitive ability used here.[42] However, unique associations of health literacy and cognitive ability with health have been reported when using a variety of different functional health literacy tests, including the REALM,[23] the TOFHLA[21 23] and the ELSA health literacy test.[22] Though attenuated, functional health literacy has also been found to have had unique associations with health after adjusting for cognitive ability created using a comprehensive test battery consisting of well-validated cognitive tests.[23] Therefore, low health literacy and poorer cognitive ability may contribute unique disadvantages in terms of navigating healthcare and looking after one's own health.[22]

This study was also interested in examining whether functional health literacy and cognitive ability were associated with reporting diabetes independent of other health-related and socioeconomic risk factors for diabetes. The largest attenuation was seen when entering health behaviours and BMI into the models. BMI and health behaviours fully attenuated the relationship between functional health literacy, cognitive ability and reporting diabetes at baseline, and partly attenuated the relationship between functional health literacy, cognitive ability and reporting diabetes during follow-up. Better cognitive ability has been associated with health promoting behaviours such as following a healthy diet and taking part in regular exercise.[4 43–45] Whereas some studies have found associations between better functional health literacy and taking part in health promoting behaviours,[46] others have not.[47] Individuals with higher functional health literacy and cognitive ability might be better equipped with the health-related skills and knowledge, and the general cognitive capabilities needed to take better care of themselves[2 48] and to follow health advice such as eating well and exercising, which, in turn, could reduce the risk of developing diabetes.[1]

Education also partly attenuated the association between functional health literacy and cognitive ability with reporting diabetes during follow-up. The association between better functional health literacy and cognitive ability with higher levels of education are well established.[7 49] Education may lead to better cognitive ability and functional health literacy, which in turn may lead to better health-related skills and lower rates of diabetes.[23] Higher cognitive ability in early life has been found to predict later educational attainment.[49] An alternative but not mutually exclusive explanation could be that higher cognitive ability may equip an individual with the skills needed to obtain higher educational qualifications. Higher educational attainment, in turn, may lead to better health (and lower risk of diabetes) by, for example, increasing health-related knowledge and decision-making skills.[23] In the current study, social class was not found to have associations with diabetes and did not appear to play an attenuating role in the association between health literacy and cognitive ability with diabetes.

This study has a number of strengths and limitations. A key strength is that it examined the association of

functional health literacy, cognitive ability and reporting diabetes longitudinally. Another strength is the relatively large sample size. One limitation is that only a subsample of participants had complete data. Those with missing data may be those with the lowest functional health literacy and cognitive ability scores. ELSA may also suffer from selective attrition such that those with increased risk of developing diabetes may be less likely to return for follow-up. The results reported here may not generalise to those with the lowest functional health literacy and/or cognitive ability. The rates of diabetes reported here do not fully match those reported in national statistics. Compared to the 2004/2005 National Diabetes Audit for England and Wales, rates of diabetes in the current study were lower for those aged 55–69 years (this study: 8.4% in men and 5.6% in women; National Diabetes Audit: approximately 10% in men, and 7% in women), but comparable in those aged 70–84 years (this study: 13.6% in men and 9.9% in women; National Diabetes Audit: approximately 13.5% in men and 10% in women).[50] Therefore, the current sample is not fully representative of people with diabetes living in England.

Another limitation is that diabetes status was self-reported. As has been shown in other ELSA studies, there is a relatively high rate of agreement between self-reported diabetes and fasting blood glucose in ELSA; however, 1.7% of participants had undiagnosed diabetes.[51] Sensitivity analysis was performed in the current study to try to identify and remove individuals with undiagnosed diabetes. Although the results were generally similar after removal of those with suspected undiagnosed diabetes, we found that health literacy was no longer associated with cross-sectional diabetes status in the subsample of participants with $HbA_{1c}$ levels. It is not clear whether these differences are due to removal of participants with suspected undiagnosed diabetes, or if it was due to bias caused by using a smaller subsample of participants who also attended the nurse interview and provided a blood sample.

The functional health literacy test used here was a brief, four-item test which had limited variance (67% of participants scored the highest score) and the psychometric properties of this measure are unknown. Although brief, this test was sensitive enough to have associations with self-reported diabetes during follow-up, and it has previously been found to have associations with mortality.[22] This brief measure only assessed functional health literacy and did not measure other components of health literacy.[8] More detailed, self-report measures of health literacy are available that assess a range of other health literacy skills, including the (self-reported) ability to access, appraise and apply health information.[52] An important next step would be to test the associations between health literacy and cognitive ability with diabetes using more detailed tests of health literacy that cover a range of other health literacy skills in addition to health-related reading comprehension.

This study found that adequate functional health literacy and higher cognitive ability were independently associated with lower odds of self-reporting diabetes at wave 2 and with reduced rates of self-reporting a new diagnosis of diabetes during a median of 9.5 years follow-up. Individuals with poor functional health literacy and/or cognitive ability might lack the health-related reading and writing skills and the general cognitive capabilities required to look after their health throughout life, which in turn, may increase the risk of being diagnosed with diabetes.

**Acknowledgements** We thank our late colleague, John Starr, for his contribution to the conception and design of this study. The authors thank the participants of the ELSA study. We are grateful to the UK Data service for supplying the data.

**Contributors** CF-R contributed to the conception and design of the project, analysed the data, interpreted the data, drafted the initial manuscript, critically revised the manuscript, and is the guarantor. JP contributed to the conception and design of the project, interpreted the data and critically revised the manuscript. ID contributed to the conception and design of the project, interpreted the data and critically revised the manuscript.

**Funding** This work was supported by the University of Edinburgh Centre for Cognitive Ageing and Cognitive Epidemiology, part of the cross council Lifelong Health and Wellbeing Initiative, funded by the Biotechnology and Biological Sciences Research Council (BBSRC), and Medical Research Council (MRC) (grant number MR/K026992/1).

**Competing interests** None declared.

**Patient and public involvement** Patients and/or the public were not involved in the design, or conduct, or reporting, or dissemination plans of this research.

**Patient consent for publication** Not applicable.

**Ethics approval** Ethical approval was obtained from the NHS Multicentre Research Ethics Committee, London (reference: MREC/01/2/91). Written informed consent was obtained from all ELSA participants. This study conformed to the principles embodied in the Declaration of Helsinki.

**Provenance and peer review** Not commissioned; externally peer reviewed.

**Data availability statement** Data may be obtained from a third party and are not publicly available. Anonymised data from the English Longitudinal Study of Ageing are available from the UK Data Service (https://https://www.ukdataservice.ac.uk/).

**ORCID iD**
Chloe Fawns-Ritchie http://orcid.org/0000-0002-7493-2228

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
