## [Reviewer comments · BMJ Open]

ARTICLE DETAILS

TITLE (PROVISIONAL)	The association of functional health literacy and cognitive ability with self-reported diabetes in the English Longitudinal Study of Ageing: A prospective cohort study
AUTHORS	Fawns-Ritchie, Chloe; Price, Jackie; Deary, Ian

VERSION 1 – REVIEW

REVIEWER	Stewart, Robert Institute of Psychiatry, Section of Epidemiology (Box 60)
REVIEW RETURNED	17-Nov-2021

GENERAL COMMENTS	This manuscript reports an analysis of ELSA cohort data in which the authors investigated cross-sectional and prospective associations of cognitive function and health literacy with diabetes prevalence and incidence. The study design is straightforward and the manuscript is clearly written. My suggestions below should be viewed as optional. 1. It might help to have a brief contextualisation of the observed diabetes prevalence and incidence in the Discussion, comparing these to other whole-population data in order to give some idea of whether the ELSA cohort is a high/low/normal risk sample. 2. Although I accept the arguments on p18-19 regarding self-report diabetes accuracy in ELSA, my slight concern is that there might be at least some people who fail to self-report their diabetes at baseline, only to recall this later. This might be particularly the case with more borderline diabetes diagnoses (e.g., diet-controlled rather than requiring medication) and clearly might associate with cognitive impairment and/or low health literacy. I wonder whether exclusion of incident cases at the first follow-up point might be worth considering as an additional sensitivity analysis.
--

REVIEWER	Eaton, Andrew University of Regina, Faculty of Social Work - Saskatoon Campus
REVIEW RETURNED	12-Jan-2022

GENERAL COMMENTS	This article reports on the association of health literacy and cognitive ability with self-reported diabetes diagnoses in a cohort of 8,669 UK participants. The article could have important findings, but constructs may be too broadly defined and undiagnosed diabetes is not sufficiently accounted for among other issues. Major Issues 1. The manuscript uses diabetes risk interchangeably with the actual
---

	measure of self-reported diabetes diagnosis from a physician. Revision is needed to how the researchers frame these two terms, as there are moderately high rates of undiagnosed diabetes. For example, the US CDC estimated that 2.8% of adults were living with undiagnosed diabetes in 2018: https://www.cdc.gov/diabetes/data/statistics-report/diagnosed-undiagnosed-diabetes.html and there are numerous studies of undiagnosed diabetes around the world, suggesting that self-reported diabetes diagnoses is not a true measure of diabetes risk. There is some discussion on this in the limitations, but the researchers seem too confident that their participants did not have undiagnosed diabetes. 2. Health literacy was assessed via participants' ability to understand information on a label for a packet of over-the-counter medication. This is only a measure of a small component of the multifaceted health literacy construct, which Sørensen et al. (2012) doi:10.1186/1471-2458-12-80 thoroughly identified. This measure may need to be renamed, and the paper thoroughly revised as a result, as 4 questions about understanding medication information does not truly constitute health literacy. Background 1. "Cognitive ability" can be a broadly interpreted term. Please provide a precise definition of how cognitive ability is conceptualized in this context, in paragraph two of the Introduction. Methods 1. Participants - As this study is focused on wave 2, it is strange that the Participants section seems dedicated to wave 1, with only a single sentence at the end mentioning waves 2-7. Recommend rewriting this section to focus specifically on the wave under study. 2. More detail about entry criteria and recruitment would be helpful. 3. Measures - what are the psychometric properties of the Health Literacy measure? 4. Analysis - there appears to be quite a bit of data manipulation peppered throughout the article with varying degrees of justification. Recommend centralizing these adjustments into a single paragraph for better assessment of this procedure. Results 1. As with the major issue stated above, we cannot know that those who did not report a diabetes diagnosis were truly without diabetes. Some nuance accounting for possible undiagnosed is needed. 2. Clearly stated frequencies of participant demographics is needed as these demographic factors are used heavily in the models. Discussion 1. It is a somewhat strange claim that 4 questions of understanding medication information can truly predict mortality. Minor issues 1. Some acronyms appear to be not spelled out at first mention. 2. What software was used for analysis?
--	--

REVIEWER	Ayre, Julie The University of Sydney Faculty of Medicine and Health, School of Public Health
REVIEW RETURNED	20-Jan-2022

GENERAL COMMENTS	This is a very thoughtful and rigorous study that looks at the
--

relationship between health literacy, cognitive ability, and diabetes. Using longitudinal data to examine the relationship between these variables is highly valuable and fills an important gap in the literature. I have some comments, mostly to help you reflect on how this research fits into the existing literature and to improve clarity about some methodological decisions.

Abstract

If possible please give some description of the health literacy measure

Please add the sample size for follow up/incident diabetes.

Introduction

- For the second paragraph (cognitive ability), it could be worth describing how cognitive ability does or doesn't differ from education, and whether the studies controlled for education.
- The relationship between diabetes and health literacy is perhaps not quite as clear as made out in the second paragraph- I have suggested an extra reference that may also be helpful. Also, similarly to cognitive ability and education, we want to know whether these associations are due to differences in socioeconomic advantage/disadvantage.

Quartuccio M, Simonsick EM, Langan S, Harris T, Sudore RL, Thorpe R, et al. The relationship of health literacy to diabetes status differs by sex in older adults. *J Diabetes Complications*. 2018;32(4):368-72. DOI: 10.1016/j.jdiacomp.2017.10.012.

- The LitCog study (Michael Wolf and colleagues) was designed specifically to observe the relationship between health literacy and cognitive performance. Consider references from this project when discussing the relationship between health literacy and cognitive ability, e.g. Wolf MS, Curtis LM, Wilson EAH, Revelle W, Waite KR, Smith SG, et al. Literacy, Cognitive Function, and Health: Results of the LitCog Study. *Journal of General Internal Medicine*. 2012;27(10):1300-7. doi: 10.1007/s11606-012-2079-4.

Methods

- Please give more detail about validation of the health literacy and cognitive measures, including references for using each of the cognitive measures and the PCA methods. Provide a rationale for why the four specific cognitive tests were selected over others.

Results

- I have no specific feedback. Presentation of results was complete and written clearly.

Discussion

- In the first paragraph there is no mention that the relationship between health literacy/cognitive ability and diabetes (cross sectional or risk) is not significant in the final model, nor is this mentioned elsewhere in the discussion.
- In the paragraph describing health literacy's variance overlapping with cognitive ability: when comparing to the literature, you could also consider how closely (or not) the health literacy tasks match the cognitive tasks. For example, the REALM and S-TOFHLA which were used in some of the references you cite in the introduction were developed as adaptations of cognitive battery tests. The NVS has a strong association with fluid abilities in other studies as well. Consider drawing on the arguments presented by Bostock and Steptoe (your reference #16), in this section of the discussion as this also uses ELSA data -- this issue is quite well described in their paper.
- It may also be worth keeping in mind that the theoretical concept of health literacy is often much broader in scope than the skills that are typically captured in performance-based health literacy instruments. As you mention in the introduction, health literacy skills as they

	pertain to diabetes can include skills for obtaining, understanding, and following health advice, and yet these kinds of skills were not necessarily measured in the current study. This is not a criticism but just a comment for further thought.  • Model 6 included social class as well as education. Currently the discussion does not mention resources and environments related to social class that may make healthy diet and regular exercise easier to achieve. Consider discussing how education and social class also afford access to resources (e.g. safe living environment, time, money, work flexibility), that might otherwise not be available. • There are already some programs such as the Diabetes Prevention Program (and adaptations) that are sensitive to low health literacy and have shown to be effective at reducing the risk of diabetes. Can you rework the final part of the discussion to more thoroughly discuss what the implications of the findings might be, or suggest a different 'future studies' direction?
--	---

VERSION 1 – AUTHOR RESPONSE

Reviewer 1

1. **It might help to have a brief contextualisation of the observed diabetes prevalence and incidence in the Discussion, comparing these to other whole-population data in order to give some idea of whether the ELSA cohort is a high/low/normal risk sample.**

Response: We thank reviewer 1 for their comments.

A section has been added to the discussion comparing the rates of diabetes in ELSA to the National Diabetes Audit statistics from 2004/05 (the baseline of the current study). Compared to these statistics, the sample used here has a slightly lower percentage of participants reporting diabetes in the 55 to 69 year age group. However, we find comparable rates of diabetes in the 70 to 84 year age group. We have now included in the discussion that a limitation of the current paper is that our sample is not fully representative of people with diabetes living in England.

Page 23: *“The rates of diabetes reported here do not fully match those reported in national statistics. Compared to the 2004/05 National Diabetes Audit for England and Wales, rates of diabetes in the current study were lower for those aged 55 to 69 years (this study: 8.4% in men and 5.6% in women; National Diabetes Audit: approximately 10% in men, and 7% in women), but comparable in those aged 70 to 84 years (this study: 13.6% in men and 9.9% in women; National Diabetes Audit: approximately 13.5% in men and 10% in women).[50] Therefore the current sample is not fully representative of people with diabetes living in England.”*

1. **Although I accept the arguments on p18-19 regarding self-report diabetes accuracy in ELSA, my slight concern is that there might be at least some people who fail to self-report their diabetes at baseline, only to recall this later. This might be particularly the case with more borderline diabetes diagnoses (e.g., diet-controlled rather than requiring medication) and clearly might associate with cognitive impairment and/or low health literacy. I wonder whether exclusion**

of incident cases at the first follow-up point might be worth considering as an additional sensitivity analysis.

Response: In ELSA, when a participant first reports a diagnosis of diabetes, they are asked the month and year they were diagnosed. Inspecting the dates of diagnosis for new diabetes cases reported at wave 3 (the first follow-up point), one individual reported that they were diagnosed with diabetes in January 1998. This participant may have forgotten to report their diagnosis at earlier waves, or they may have been erroneously classified as not having diabetes by the interviewer. All other newly diagnosed cases at wave 3 report being diagnosed with diabetes on or after August 2003, suggesting that these participants were indeed diagnosed after the wave 2 assessment and did not forget at earlier waves.

As part of this revision, we have now carried out sensitivity analyses to test whether the results reported in the main text change after removing individuals with undiagnosed diabetes (here defined as not self-reporting diabetes but with a Hba1c level of ≥ 47.5 mmol/mol (6.5%)). Removal of these participants could also remove any participants that may have forgotten they had diabetes because they were controlling their diabetes not using medication.

Generally speaking, the results of the sensitivity analyses are similar to those reported in the main text, but there are a few differences. For the cross-sectional analyses ($n=5,671$; 399 with diabetes; see Supplementary Table S5), the associations between cognitive ability and diabetes status at wave 2 are very similar to those reported in the main model. The size of the associations between functional health literacy and diabetes were reduced and were no longer significant.

For the Cox models ($n=4,206$; 212 reporting diabetes during follow-up; Supplementary Table S6), the size of the associations between limited functional health literacy and self-reporting diabetes during follow-up became even stronger. In the fully-adjusted model, the association between limited functional health literacy and diabetes remained significant. For cognitive ability, the strength of the associations were generally similar to the main models. However, after adjusting for BMI and health behaviours (Model 4) the size of the association between cognitive ability and diabetes is slightly attenuated and no longer significant.

Reviewer 2

- 1. The manuscript uses diabetes risk interchangeably with the actual measure of self-reported diabetes diagnosis from a physician. Revision is needed to how the researchers frame these two terms, as there are moderately high rates of undiagnosed diabetes. For example, the US CDC estimated that 2.8% of adults were living with undiagnosed diabetes in 2018: <https://www.cdc.gov/diabetes/data/statistics-report/diagnosed-undiagnosed-diabetes.html> and there are numerous studies of undiagnosed diabetes around the world, suggesting that self-reported diabetes diagnoses is not a true measure of diabetes risk. There is some discussion on this in the limitations, but the researchers seem too confident that their participants did not have undiagnosed diabetes.**

Response: We thank reviewer 2 for their comments. We have updated the manuscript to incorporate their suggestions and we think this has helped to improve the clarity of the paper.

We have revised the manuscript to make it clear that we are looking at the association with self-reported doctor diagnosis of diabetes and not a true measure of diabetes risk. For example:

Page 7: *“Second, participants without diabetes at baseline were followed-up for up to 10 years to determine whether functional health literacy and cognitive ability were independently associated with newly reporting diabetes during follow-up.”*

Page: 12: *“. Cox regression was used to investigate whether functional health literacy and cognitive ability test scores at wave 2 were associated with newly reported diabetes between waves 2 and 7.”*

Page 17: *“The HRs and 95% CIs for the association between functional health literacy, cognitive ability and self-reporting diabetes during follow-up are reported in Table 4 (and Supplementary Table S2).”*

Page 20: *“Adequate health literacy and better cognitive ability, measured at wave 2, were associated with reduced risk of self-reporting diabetes during a median of 9.5 years follow-up.”*

We have also carried out sensitivity analyses to understand whether the results reported in the main text change after removing individuals with possible undiagnosed diabetes. A sub-sample of participants have HbA1c levels available from the nurse interview conducted at wave 2, 4, and 6. Here, we have categorised individuals who do not self-report diabetes but who have HbA1c levels ≥ 7.5 mmol/mol (6.5%) as having possible undiagnosed diabetes. We have re-run the models reported in the main text on the sub-sample of participants with HbA1c levels, after removal of participants with suspected undiagnosed diabetes.

We removed participants categorised with undiagnosed diabetes because the paper focuses on self-reported doctor diagnosed diabetes and we felt that people with undiagnosed diabetes are likely to differ from those who been diagnosed with diabetes. Therefore we did not combine participants with undiagnosed diabetes and self-reported doctor diagnosis of diabetes into one “diabetes” group.

Generally speaking, the results of the sensitivity analyses are similar to those reported in the main text, but there are a few differences. For the cross-sectional analyses (Supplementary Table S5), the associations between cognitive ability and diabetes status at wave 2 are very similar to those reported in the main models. The size of the associations between functional health literacy and diabetes were reduced and were no longer significant.

For the Cox models (Supplementary Table S6), the size of the associations between limited functional health literacy and self-reporting diabetes during follow-up became even stronger. In the fully-adjusted model, the association between limited functional health literacy and diabetes remained significant. For cognitive ability, the strength of the associations were generally similar to the main models. However, after adjusting for BMI and health behaviours (Model 4) the size of the association between cognitive ability and diabetes is slightly attenuated and no longer significant.

- 1. Health literacy was assessed via participants' ability to understand information on a label for a packet of over-the-counter medication. This is only a measure of a small component of the multifaceted health literacy construct, which Sørensen et al. (2012) doi:10.1186/1471-2458-12-80 thoroughly identified. This measure may need to be renamed, and the paper thoroughly revised as a result, as 4 questions about understanding medication information does not truly constitute health literacy.**

Response: We agree that health literacy is a multifaceted construct and the brief test used in the current study only assesses a relatively small part of this construct. We have now updated the paper to refer to this assessment as a measure of “functional health literacy” – the health-

related reading, writing and numeracy skills needed to understand health information. We have now updated the title of the manuscript to include “functional health literacy”.

We have also added additional text to the introduction to highlight that health literacy is multifaceted and that the measure of health literacy used here is assessing only one small part of that construct:

Pages 5-6: *“Health literacy is the “capacity to obtain, process and understand basic health information and services needed to make basic health decisions”[7], and it might also play a role in diabetes. Health literacy is a multifaceted construct thought to encompass all of the skills required to make decisions about one’s health, including the ability to access, appraise and apply health information.[8, 9] One component of health literacy is functional health literacy – the reading, writing and numeracy skills needed to understand basic health information.[10] These skills are thought to be required, for example, to understand and correctly follow the instructions on a packet of prescription medication.”*

We have also updated the discussion to highlight that using a brief test of functional health literacy does not adequately assess the multifaceted construct of health literacy.

Page 24: *“This brief measure only assessed functional health literacy and did not measure other components of health literacy.[8] More detailed, self-report measures of health literacy are available that assess a range of other health literacy skills, including the (self-reported) ability to access, appraise and apply health information.[52] An important next step would be to test the associations between health literacy and cognitive ability with diabetes using more detailed tests of health literacy that cover a range of other health literacy skills in addition to health-related reading comprehension.”*

1. **"Cognitive ability" can be a broadly interpreted term. Please provide a precise definition of how cognitive ability is conceptualized in this context, in paragraph two of the Introduction.**

Response: Here, we are defining cognitive ability as the composite term for a wide range of mental capabilities including the ability to learn, plan process information and solve problems. We have updated the introduction to include this information.

In this paragraph, we also include information about how cognitive ability and education are related but distinct from each other. We have included this at the request of Reviewer 3.

Page 5: *“Cognitive ability can be conceptualised as a composite term for a range of different but overlapping mental capabilities, including the ability to learn, plan, problem solve and process information.[2] Cognitive ability is closely related to but distinct from educational attainment and correlations between cognitive ability and education range from 0.40 to 0.80.[3] This general mental capability has been found to be associated with many different aspects of health.[2]”*

1. **Participants - As this study is focused on wave 2, it is strange that the Participants section seems dedicated to wave 1, with only a single sentence at the end mentioning waves 2-7. Recommend rewriting this section to focus specifically on the wave under study.**

Response: The participant section focuses on wave 1 because this is when the participants used in the current study were recruited. We feel it's important to include this information to provide the reader enough information to understand how the current sample were recruited. We have therefore left this information in manuscript. We now provide more information about wave 2 and we also make it clear when describing the assessments that participant's completed took place at every wave and not just at wave 1.

Page 8: *"This study used data from core members of the ELSA study, a prospective cohort study of community-dwelling adults residing in England. ELSA was designed to be representative of adults aged 50 years and older living in England.[25] The wave 1 (2002-03) sample consisted of 11,391 participants who had previously participated in the Health Survey for England between 1998 and 2001, who were born before 1 March 1952, and who were living in a private household in England.[25] ELSA participants have been followed up every two years and the sample has been refreshed at waves 3, 4, 6 and 7 to ensure the sample is representative of adults aged over 50 years. The present study used data from waves 2 (2004-05) to 7 (2014-15), and baseline, here, was considered to be wave 2 (n=8,726), which was when the functional health literacy assessment was introduced.*

At each wave, a face-to-face interview was used to measure topics including health, lifestyle and economic circumstances. Face-to-face interviews were carried out in the participant's own home using computer-assisted interviewing. Participants answered a self-completion questionnaire including questions about diet and alcohol consumption. A nurse interview was carried out at waves 2, 4 and 6 to assess physical measurements including height and weight, and blood and saliva samples were taken to measure biomarkers of disease. Detailed descriptions of the sample design and data collected in ELSA are reported elsewhere.[25]"

1. More detail about entry criteria and recruitment would be helpful.

Response: We have now provided additional details in the methods section about the entry criteria and recruitment. As above, this focuses on wave 1, as this was when participants in the current study were recruited.

Pages 8: *"This study used data from core members of the ELSA study, a prospective cohort study of community-dwelling adults residing in England. ELSA was designed to be representative of adults aged 50 years and older living in England.[25] The wave 1 (2002-03) sample consisted of 11,391 participants who had previously participated in the Health Survey for England between 1998 and 2001, who were born before 1 March 1952, and who were living in a private household in England.[25] ELSA participants have been followed up every two years and the sample has been refreshed at waves 3, 4, 6 and 7 to ensure the sample is representative of adults aged over 50 years. The present study used data from waves 2 (2004-05) to 7 (2014-15), and baseline, here, was considered to be wave 2 (n=8,726), which was when the functional health literacy assessment was introduced."*

1. Measures - what are the psychometric properties of the Health Literacy measure?

Response: The health literacy items used in ELSA were taken from the Adult Literacy and Life Skills Survey (Clouston, Manganello, & Richards, 2017) and the International Adult Literacy Survey (Kirsch, 2001). We have now provided additional information in the methods about the source of the ELSA health literacy questions:

Page 9: “A 4-item functional health literacy test taken from the Adult Literacy and Life Skills Survey,[26] and the International Adult Literacy Survey[27] was administered during the wave 2 interview.

From inspection of these reports, it appears that psychometric evaluations were carried out during the development of these items, however, the psychometric properties for the items used in the ELSA study not reported. This is a limitation and we have updated the discussion to include this limitation.

Page 24: “The functional health literacy test used here was a brief, four-item test which had limited variance (67% of participants scored the highest score) and the psychometric properties of this measure are unknown.”

The ELSA health literacy test is similar to other well-validated measures of functional health literacy including the Test of Functional Health Literacy In Adults (Parker, Baker, Williams, & Nurss, 1995), which also asks participants to read and answer questions of a prescription label. The reported associations between the ELSA health literacy test and different aspects of health also add support to its construct validity.

References

- Clouston, S. A. P., Manganello, J. A., & Richards, M. (2017). A life course approach to health literacy: the role of gender, educational attainment and lifetime cognitive capability. *Age and Ageing*, 46(3), 493-499. doi:10.1093/ageing/afw229
- Kirsch, I. S. (2001). The International Adult Literacy Survey (IALS): Understanding What is Measured. 2001(2), i-61. doi:<https://doi.org/10.1002/j.2333-8504.2001.tb01867.x>
- Parker, R. M., Baker, D. W., Williams, M. V., & Nurss, J. R. (1995). The test of functional health literacy in adults: A new instrument for measuring patients' literacy skills. *Journal of General Internal Medicine*, 10(10), 537-541. doi:10.1007/bf02640361

- 1. Analysis - there appears to be quite a bit of data manipulation peppered throughout the article with varying degrees of justification. Recommend centralizing these adjustments into a single paragraph for better assessment of this procedure.**

Response: We think that by “data manipulation” you are referring to where we have removed certain responses from the cognitive test scores before entering these into a principal component analysis, and where we have made changes to models after checking the model assumptions. For the cognitive tests scores, these decisions were made following inspection of the data and when preparing to create a measure of general cognitive ability. Changes to the model were made after checking the assumptions. To make it explicitly clear to the reader why these decisions were made, we think the best place to document these changes in the manuscript is in the relevant sections that they were carried out (i.e., when creating the general measure of cognitive ability, and when reporting the models).

We have however tried to provide more detail as to why these decisions were made.

Pages 10-11: “Scores of 0 on animal fluency (n=48) and letter cancellation (n=3) were removed as scores of 0 on these tests suggest participants either did not complete the task or did not understand the task. Scores of ≥ 50 on animal fluency (n=4), and ≥ 60 on the letter cancellation (n=3) were removed as these scores were extremely high given the 60 second time limit for these tests and these values are greater than 4 SDs from the mean.”

Page 14-15: “Box-Tidwell tests were performed whereby an interaction term between each continuous predictor variable and the log of that variable were added to the model to check the assumption that there is a linear relationship between each continuous predictor and the logit of the outcome. The interaction between age and log(age) and the interaction between

number of cardiovascular comorbidities and log(number of cardiovascular comorbidities) was significant. Therefore the assumptions of the linearity of the logit was violated. To overcome this, an age-squared term was included in all models, and a squared term for number of cardiovascular comorbidities was included in Models 5 and 7.

Some data manipulations were outwith our control. For example, participants aged over 90 years had their age set to 90 prior to the data being released to us. The text has been edited to make it clear to the reader this was done before we were provided the ELSA data.

Page 11: *“Prior to releasing data, ELSA set the age of all participants aged over 90 years to 90 years to reduce the risk of disclosure.”*

- 1. As with the major issue stated above, we cannot know that those who did not report a diabetes diagnosis were truly without diabetes. Some nuance accounting for possible undiagnosed is needed.**

Response: As described in point 1 above, we have now tried to identify participants with undiagnosed diabetes and perform sensitivity analyses removing these individuals.

A sub-sample of the participants used in the current study have HbA1c levels available from the nurse interview conducted at wave 2, 4, and 6. Here, we have categorised individuals who do not self-report diabetes but who have HbA1c levels ≥ 47.5 mmol/mol (6.5%) as having possible undiagnosed diabetes. We have re-run the models reported in the main text on the sub-sample of participants with HbA1c levels, after removal of participants with suspected undiagnosed diabetes.

We removed participants categorised with undiagnosed diabetes because the paper focuses on self-reported doctor diagnosed diabetes and we felt that people with undiagnosed diabetes are likely to differ from those who have been diagnosed with diabetes. Therefore we did not combine those with undiagnosed diabetes and with a self-reported doctor diagnosis of diabetes into one “diabetes” group.

Generally speaking, the results of the sensitivity analyses are similar to those reported in the main text, but there are a few differences. For the cross-sectional analyses (Supplementary Table S5), the associations between cognitive ability and diabetes status at wave 2 are very similar to those reported in the main models. The size of the associations between functional health literacy and diabetes were reduced and were no longer significant.

For the Cox models (Supplementary Table S6), the size of the associations between limited functional health literacy and self-reporting diabetes during follow-up became even stronger. In the fully-adjusted model, the association between limited functional health literacy and diabetes remained significant. For cognitive ability, the strength of the associations were generally similar to the main models. However, after adjusting for BMI and health behaviours (Model 4) the size of the association between cognitive ability and diabetes is slightly attenuated and no longer significant.

We note in the discussion that any differences between the results reported in the main text and in the sensitivity analyses could also be due to the fact that this analysis was performed using a smaller sub-sample of participants who attended the nurse interview and who provided a blood sample.

Page 24: *“Sensitivity analysis was performed in the current study to try to identify and remove individuals with undiagnosed diabetes. Although the results were generally similar after removal of those with suspected undiagnosed diabetes, we found that health literacy was no longer associated with cross-sectional diabetes status in the sub-sample of participants with HbA1c levels. It is not clear whether these differences are due to removal of participants with*

suspected undiagnosed diabetes, or if it was due to bias caused by using a smaller sub-sample of participants who also attended the nurse interview and provided a blood sample.”

- 1. Clearly stated frequencies of participant demographics is needed as these demographic factors are used heavily in the models.**

Response: We have now added age and sex frequencies to the main text as these covariates are included in all models. We have not added the frequencies of the remainder of the demographic variables as these are all presented in Table 1 which will be on or close to the page reporting the results. After incorporating the changes requested by the reviewers, the manuscript is now much longer than it was before. We want to limit the length and feel that adding the frequencies for all of the demographics would be lengthy and redundant as all of these frequencies are reported in Table 1.

We have added “(Table 1)” at the end of the describing the demographics to remind readers these frequencies are reported in Table 1.

Page 14: *“Participants with diabetes were older (diabetes mean=69.36, SD=9.16; no diabetes mean=66.46, SD=9.70) and more likely to be male (53.5% versus 44.2%) than those without. Those with diabetes were also more likely to leave full-time education at a younger age, be from a less professional social class, have a higher BMI, consume less alcohol, be inactive, and self-report more cardiovascular comorbidities (Table 1).”*

Page 16: *“Participants who reported diabetes were younger (diabetes mean=65.51, SD=8.59; no diabetes mean=66.0; SD=9.53) and more likely to be male (51.8% versus 43.2%) than those without. Compared to those without diabetes, participants who reported diabetes during follow-up were more likely to have left full-time education at a younger age, be from a less professional social class, smoke, consume less alcohol, be inactive, and to report more cardiovascular comorbidities at wave 2 (Table 1).”*

- 1. It is a somewhat strange claim that 4 questions of understanding medication information can truly predict mortality.**

Response: We have edited the discussion to state that this test has been found to “have associated with mortality” instead of suggesting it truly predicts mortality.

Page 24: *“Although brief, this test was sensitive enough to have associations with self-reported diabetes during follow-up, and it has previously been found to have associations with mortality.[22]”*

- 1. Some acronyms appear to be not spelled out at first mention**

Response: We have gone through the manuscript and now spell out acronyms on first mention.

- 1. What software was used for analysis?**

Response: R statistical software was used for all analyses. This is now reported in the methods section.

Page 12: "All analyses was performed in R."

Reviewer 3

- 1. Abstract: If possible please give some description of the health literacy measure. Please add the sample size for follow up/incident diabetes.**

Response: We thank reviewer 3 for their helpful comments. We have incorporated these changes into the manuscript and we think these changes have improved the manuscript.

The abstract has now been updated to include a brief description of the health literacy measure (as well as more information about the cognitive tests), and we now include the sample size for the longitudinal analyses.

Page 2: "8,669 ELSA participants (mean age=66.7, SD=9.7) who completed a brief functional health literacy test assessing health-related reading comprehension, and 4 cognitive tests assessing declarative memory, processing speed and executive function at wave 2."

Page 2: "Cox regression was used to test the associations of functional health literacy and cognitive ability measured at wave 2 with self-reporting diabetes over a median of 9.5 years follow-up (n=6,961)."

- 1. Introduction: For the second paragraph (cognitive ability), it could be worth describing how cognitive ability does or doesn't differ from education, and whether the studies controlled for education.**

Response: We have now updated the introduction to include a sentence highlighting that cognitive ability and education are related but distinct with correlations of between .40 and .80 being reported.

Page 5: "Lower cognitive ability may be a risk factor for diabetes. Cognitive ability can be conceptualised as a composite term for a range of different but overlapping mental capabilities, including the ability to learn, plan, problem solve and process information.[2] Cognitive ability is closely related to but distinct from educational attainment and correlations between cognitive ability and education range from 0.40 to 0.80.[3]"

When discussing the studies looking at the association between cognitive ability and diabetes, we now report whether they adjusted for education (and other sociodemographic variables).

Page 5: "One study[4] found that childhood cognitive ability did not predict diabetes in midlife when individually adjusting for a range of demographic variables including education. Others have found that lower cognitive ability in early life was associated with higher risk of diabetes in adulthood.[5, 6] Whereas the first study[5] did not adjust for educational attainment or measures of socioeconomic status, the latter[6] found that individuals with lower cognitive ability in early adulthood had higher rates of diabetes in midlife, even after adjusting for education and indicators of socioeconomic status."

1. **The relationship between diabetes and health literacy is perhaps not quite as clear as made out in the second paragraph- I have suggested an extra reference that may also be helpful. Also, similarly to cognitive ability and education, we want to know whether these associations are due to differences in socioeconomic advantage/disadvantage.**

Quartuccio M, Simonsick EM, Langan S, Harris T, Sudore RL, Thorpe R, et al. The relationship of health literacy to diabetes status differs by sex in older adults. J Diabetes Complications. 2018;32(4):368-72. DOI: 10.1016/j.jdiacomp.2017.10.012.

Response: Thank you for highlighting this paper to us. The paragraph detailing previous research on the association between health literacy and diabetes has now been updated to include this study.

Page 6: *“Associations between health literacy and diabetes may differ by sex. Women with low health literacy were found to be more than twice as likely to have diabetes compared to those with high literacy after adjusting for age, race, income, education, body mass index (BMI), and smoking and alcohol status, however, health literacy was not associated with diabetes status in men.[13] Individuals with lower functional health literacy – at least in women – might lack the health-related reading and writing skills required to obtain, understand and follow health advice, such as eating well and exercising, which might reduce the risk of diabetes.[7]”*

We have also provided additional details about whether the studies discussed have adjusted for indicators of socioeconomic status.

Page 6: *“In cross-sectional studies, rates of diabetes are higher in those with low functional health literacy, even after adjusting for age, sex, income and education.[11, 12] In one study, participants with inadequate functional health literacy were 48% more likely to report having diabetes when compared to participants with adequate health literacy, adjusting for sociodemographic and health variables.[12]”*

1. **The LitCog study (Michael Wolf and colleagues) was designed specifically to observe the relationship between health literacy and cognitive performance. Consider references from this project when discussing the relationship between health literacy and cognitive ability, e.g. Wolf MS, Curtis LM, Wilson EAH, Revelle W, Waite KR, Smith SG, et al. Literacy, Cognitive Function, and Health: Results of the LitCog Study. Journal of General Internal Medicine. 2012;27(10):1300-7. doi: 10.1007/s11606-012-2079-4.**

Response: We have now included the Wolf et al (2012) paper in the references for papers showing correlations between functional health literacy and cognitive ability.

Page 7: *“Functional health literacy and cognitive ability test scores are positively correlated.[17-19]”*

We also now describe the results of the of Wolf et al (2012) paper in the paragraph describing the relationship between health literacy and cognitive ability.

Page 7: *“One study[19] sought to determine whether health literacy and cognitive ability had independent associations with performance on various health-related tasks, including comprehending written and video-presented health information and using health-related*

props, such as a pill bottle. Using three different measures of functional health literacy, the association between functional health literacy and performance on the health-related tasks were attenuated by between 70.6% and 77.7% when including cognitive ability in the same model compared to models not including cognitive ability.[19] Any association between functional health literacy and diabetes may be attenuated when also measuring cognitive ability.”

- 1. Methods: Please give more detail about validation of the health literacy and cognitive measures, including references for using each of the cognitive measures and the PCA methods. Provide a rationale for why the four specific cognitive tests were selected over others.**

Response: The health literacy items used in ELSA were taken from the Adult Literacy and Life Skills Survey (Clouston et al., 2017) and the International Adult Literacy Survey (Kirsch, 2001).e have now provided additional information in the methods about the source of the ELSA health literacy questions:

Page 9: *“A 4-item functional health literacy test taken from the Adult Literacy and Life Skills Survey,[26] and the International Adult Literacy Survey[27] was administered during the wave 2 interview.”*

From inspection of these reports, it appears that a psychometric evaluations were carried out during the development of these items, however, the psychometric properties for the items used in the ELSA study are not reported. This is a limitation and we have updated the discussion to include this limitation.

Page 24: *“The functional health literacy test used here was a brief, four-item test which had limited variance (67% of participants scored the highest score) and the psychometric properties of this measure are unknown.”*

However, the ELSA health literacy test is similar to other well-validated measures of health literacy including the Test of Functional Health Literacy In Adults (Parker et al., 1995), which also asks participants to read a prescription label and answer questions about the information on the prescription label. The reported associations between the ELSA health literacy test and different aspects of health also add support to its construct validity.

We are not aware of any reports from ELSA on the psychometric properties of the cognitive assessment used in ELSA. However, the cognitive tests included here are commonly used cognitive tests designed to assess cognitive abilities that decline with increasing age. We have included more information about what cognitive domains each test ae assessing and how the tests used in ELSA are the same as or similar to other validated and well-used cognitive tests.

Page 10: *“Word list learning tests, in which participants are required to remember a list of words immediately and then after a delay are commonly used to assess verbal declarative memory and learning.[33] Here, the immediate and delayed word recall tests were used. Participants were read a list of 10 words and were asked to immediately recall as many of the words as possible. The score was the number of words recalled immediately. After a short delay, in which the words were not repeated, participants were asked to remember the 10 words again. The score was the number of words recalled after a delay. Verbal fluency tests, in which participants are asked to produce as many words as possible in a set time following a set of rules, are often used to measure executive function.[33] Category fluency was used to assess executive function in ELSA. Participants were instructed to name as many animals as possible. The score was the number of animals named in 60 seconds. Tests of processing*

speed involve completing a simple task as quickly as possible and common tests include using a code to write as many symbols as possible, or finding symbols amongst distractors and scoring them out as quickly as possible.[33, 34] Letter cancellation was used to assess processing speed. Participants were presented with a piece of paper containing letters of the alphabet arranged in rows and columns. The task was to scan the piece of paper and score out all Ps and Ws. The score was the combined number of Ps and Ws scored out in 60 seconds.”

We now also provide more information about the use of principal component analysis to make a measure of general cognitive ability:

Page 10: “Scores on different cognitive tests tend to be positively correlated.[30] Data reduction techniques such as principal component analysis (PCA) are often used to capture the covariance among a range of difference cognitive tests. This shared variance can then be used as a measure of general cognitive ability.[31] Four tests administered during the wave 2 interview that are designed to assess cognitive domains that decline with increasing age[32] were entered into a PCA to create a measure general cognitive ability.”

We have updated the methods section to detail why certain cognitive tests administered at wave 2 were not included in the principal components analysis.

Page 11: “We did not include tests of self-reported memory, prospective memory or orientation in time in the measure of general cognitive ability. Self-reported memory was not included because this is a subjective test. Prospective memory was not included because the test consists of only one trial. Orientation in time is a four item test in which participants are asked to recall the date. It has limited variance and is most frequently used as a brief screening tool for cognitive impairment.”

References

- Clouston, S. A. P., Manganello, J. A., & Richards, M. (2017). A life course approach to health literacy: the role of gender, educational attainment and lifetime cognitive capability. *Age and Ageing*, 46(3), 493-499. doi:10.1093/ageing/afw229
- Kirsch, I. S. (2001). The International Adult Literacy Survey (IALS): Understanding What is Measured. 2001(2), i-61. doi:<https://doi.org/10.1002/j.2333-8504.2001.tb01867.x>
- Parker, R. M., Baker, D. W., Williams, M. V., & Nurss, J. R. (1995). The test of functional health literacy in adults: A new instrument for measuring patients' literacy skills. *Journal of General Internal Medicine*, 10(10), 537-541. doi:10.1007/bf02640361

- 1. Discussion In the first paragraph there is no mention that the relationship between health literacy/cognitive ability and diabetes (cross sectional or risk) is not significant in the final model, nor is this mentioned elsewhere in the discussion.**

Response: We have now updated the start of the discussion section to more clearly report how the associations between health literacy and cognitive ability with diabetes change when adjusting for the different sets of covariates examined in the current study. We also make it clear in the discussion that neither health literacy nor cognitive ability were associated with diabetes (at wave 2 or during follow-up) when all covariates were entered in the same model.

Page 20: “Using a sample of middle-aged and older adults living in England, the present study found that adequate functional health literacy and better cognitive ability were associated with lower odds of self-reporting diabetes. These associations were attenuated when functional health literacy and cognitive ability were entered in the same model, though both independently contributed to diabetes. These associations were further attenuated and

non-significant when adjusting for BMI and health behaviours. Adjusting for cardiovascular comorbidities and indicators of socioeconomic status did not attenuate the association between cognitive ability and diabetes, however, for functional health literacy there was a small attenuation and these associations were no longer significant. When adjusting for all covariates simultaneously, neither functional health literacy nor cognitive ability was associated with diabetes at wave 2.

Adequate health literacy and better cognitive ability, measured at wave 2, were associated with reduced risk of self-reporting diabetes during a median of 9.5 years follow-up. Both functional health literacy and cognitive ability were independently associated with self-reported diabetes when both were entered in the same model. These associations remained when separately adjusting for BMI and health behaviours, cardiovascular comorbidities, and education and social class. However, neither health literacy nor cognitive ability were associated with reporting diabetes during follow-up when all covariates were entered together.”

- 1. In the paragraph describing health literacy’s variance overlapping with cognitive ability: when comparing to the literature, you could also consider how closely (or not) the health literacy tasks match the cognitive tasks. For example, the REALM and S-TOFHLA which were used in some of the references you cite in the introduction were developed as adaptations of cognitive battery tests. The NVS has a strong association with fluid abilities in other studies as well. Consider drawing on the arguments presented by Bostock and Steptoe (your reference #16), in this section of the discussion as this also uses ELSA data -- this issue is quite well described in their paper.**

Response: Studies have found that health literacy and cognitive ability have unique associations with health when using a variety of different health literacy tests (the ELSA health literacy test, the REALM, NVS and the TOFHLA) as well as using a variety of different cognitive assessments (from brief screening tools to much more detailed cognitive assessments). These associations do not seem to be dependent on any one type of health literacy test being more closely related to cognitive function driving this association.

We have updated the discussion to describe some of the similarities between tests of cognitive ability and health literacy.

Page 21: *“There are obvious similarities between tests of cognitive ability and functional health literacy. The Rapid Estimate of Adult Literacy in Medicine (REALM)[39] is a popular health literacy test which involves the ability to read and pronounce health-related words of varying complexity. More ecologically valid assessments of functional health literacy such as the Test of Functional Health Literacy in Adults (TOFHLA)[10] and the health literacy test used in the current study involve participants using mock health-related props, such as prescription labels or a medical appointment slips, and answering questions about the information presented. Successful completion of these tests will require the ability to process information, plan and problem solve (i.e., cognitive ability).[2]”*

We also now discuss cognitive ability and health literacy tests used in studies that have found that health literacy and cognitive ability both independently contribute to health (i.e., the argument made by Bostock and Steptoe).

Page 21: *“The level of independence between health literacy and cognitive ability may vary depending on the assessments used to measure health literacy and cognitive ability.[22]”*

Pages 21-22: *“However, unique associations of health literacy and cognitive ability with health have been reported when using a variety of different functional health literacy tests,*

including the REALM[23], the TOFHLA[21, 23] and the ELSA health literacy test[22]. Though attenuated, functional health literacy has also been found to have had unique associations with health after adjusting for cognitive ability created using a comprehensive test battery consisting of well-validated cognitive tests[23]. Therefore, low health literacy and poorer cognitive ability may contribute unquidisadvantages in terms of navigating healthcare and looking after one's own health.[22]"

- 1. It may also be worth keeping in mind that the theoretical concept of health literacy is often much broader in scope than the skills that are typically captured in performance-based health literacy instruments. As you mention in the introduction, health literacy skills as they pertain to diabetes can include skills for obtaining, understanding, and following health advice, and yet these kinds of skills were not necessarily measured in the current study. This is not a criticism but just a comment for further thought.**

Response: We acknowledge that the current study only assesses functional health literacy (the health-related reading and writing skills required to process health information) and that the construct of health literacy is much broader than this. We have updated the manuscript to refer to "functional health literacy" through instead of "health literacy". We also describe the health literacy and functional health literacy in more detail in the introduction.

Pages 5-6: *"Health literacy is the "capacity to obtain, process and understand basic health information and services needed to make basic health decisions"[7], and it might also play a role in diabetes. Health literacy is a multifaceted construct thought to encompass all of the skills required to make decisions about one's health, including the ability to access, appraise and apply health information.[8, 9] One component of health literacy is functional health literacy – the reading, writing and numeracy skills needed to understand basic health information.[10] These skills are thought to be required, for example, to understand and correctly follow the instructions on a packet of prescription medication."*

And we include in the discussion that a limitation of the current study is that we have only assessed one small component of health literacy (functional health literacy) and we did not investigate the construct as a whole.

Page 24: *"This brief measure only assessed functional health literacy and did not measure other components of health literacy.[8] More detailed, self-report measures of health literacy are available that assess a range of other health literacy skills, including the (self-reported) ability to access, appraise and apply health information.[52] An important next step would be to test the associations between health literacy and cognitive ability with diabetes using more detailed tests of health literacy that cover a range of other health literacy skills in addition to health-related reading comprehension."*

- 1. Model 6 included social class as well as education. Currently the discussion does not mention resources and environments related to social class that may make healthy diet and regular exercise easier achieve. Consider discussing how education and social class also afford access to resources (e.g. safe living environment, time, money, work flexibility), that might otherwise not be available.**

Response: We only discussed the attenuating effect of education in relation to the association between health literacy and cognitive ability with risk of diabetes at follow-up because when adding education and social class to the models education was associated

with risk of developing diabetes during follow-up (but not with cross-sectional diabetes status). Social class was not associated with cross-sectional diabetes status at wave 2 (Supplementary Table S1) or with risk of diabetes during follow-up (Supplementary Table S2) and therefore is unlikely to play an attenuating role in the association between health literacy, cognitive ability and diabetes.

We have now added a sentence to the discussion to highlight to the reader that in our study at least, social class was not associated with diabetes and did not appear to play an attenuating role in the association between health literacy, cognitive ability and diabetes.

Page 23: *“In the current study, social class was not found to have associations with diabetes and did not appear to play an attenuating role in the association between health literacy and cognitive ability with diabetes.”*

1. **There are already some programs such as the Diabetes Prevention Program (and adaptations) that are sensitive to low health literacy and have shown to be effective at reducing the risk of diabetes. Can you rework the final part of the discussion to more thoroughly discuss what the implications of the findings might be, or suggest a different 'future studies' direction?**

Response: We have re-worked the final two paragraphs so that the second last paragraph details what future studies we think should be conducted. Here we have said that we think studies that use tests of health literacy that more accurately capture the multifaceted nature of health literacy are needed. The final paragraph now concludes with a summary of the findings and general statement that those with low health literacy and cognitive ability might lack the health-related reading and writing skills, and the general abilities to learn, plan and problem solve that are required to look after one’s health.

Pages 24-25: *“This brief measure only assessed functional health literacy and did not measure other components of health literacy.[8] More detailed, self-report measures of health literacy are available that assess a range of other health literacy skills, including the (self-reported) ability to access, appraise and apply health information.[52] An important next step would be to test the associations between health literacy and cognitive ability with diabetes using more detailed tests of health literacy that cover a range of other health literacy skills in addition to health-related reading comprehension.*

This study found that adequate functional health literacy and higher cognitive ability were independently associated with lower odds of self-reporting diabetes at wave 2 and with reduced rates of self-reporting a new diagnosis of diabetes during a median of 9.5 years follow-up. Individuals with poor functional health literacy and/or cognitive ability might lack the health-related reading and writing skills and the general cognitive capabilities required to look after their health throughout life, which in turn, may increase the risk of being diagnosed with diabetes.”

VERSION 2 – REVIEW

REVIEWER	Eaton, Andrew University of Regina, Faculty of Social Work - Saskatoon Campus
REVIEW RETURNED	29-Mar-2022
GENERAL COMMENTS	The authors have addressed my concerns and the manuscript is much improved.

REVIEWER	Ayre, Julie The University of Sydney Faculty of Medicine and Health, School of Public Health
REVIEW RETURNED	24-Mar-2022
GENERAL COMMENTS	Thank you, all of my comments have been addressed.